# The Intestinal Microbiome after Traumatic Injury

**DOI:** 10.3390/microorganisms11081990

**Published:** 2023-08-02

**Authors:** Jennifer A. Munley, Stacey L. Kirkpatrick, Gwendolyn S. Gillies, Letitia E. Bible, Philip A. Efron, Ravinder Nagpal, Alicia M. Mohr

**Affiliations:** 1Sepsis and Critical Illness Research Center, Department of Surgery, University of Florida College of Medicine, Gainesville, FL 32610, USA; jennifer.munley@surgery.ufl.edu (J.A.M.); stacey.kirkpatrick@surgery.ufl.edu (S.L.K.); gwendolyn.gillies@surgery.ufl.edu (G.S.G.); letitia.bible@surgery.ufl.edu (L.E.B.); philip.efron@surgery.ufl.edu (P.A.E.); 2Department of Nutrition & Integrative Physiology, Florida State University College of Health and Human Sciences, Tallahassee, FL 32306, USA; rnagpal@fsu.edu

**Keywords:** trauma, injury, hemorrhagic shock, microbiome, dysbiosis, pathobiome

## Abstract

The intestinal microbiome plays a critical role in host immune function and homeostasis. Patients suffering from—as well as models representing—multiple traumatic injuries, isolated organ system trauma, and various severities of traumatic injury have been studied as an area of interest in the dysregulation of immune function and systemic inflammation which occur after trauma. These studies also demonstrate changes in gut microbiome diversity and even microbial composition, with a transition to a pathobiome state. In addition, sex has been identified as a biological variable influencing alterations in the microbiome after trauma. Therapeutics such as fecal transplantation have been utilized to ameliorate not only these microbiome changes but may also play a role in recovery postinjury. This review summarizes the alterations in the gut microbiome that occur postinjury, either in isolated injury or multiple injuries, along with proposed mechanisms for these changes and future directions for the field.

## 1. Microbiome

The microbiome is a complex system of microbial organisms that contributes to homeostasis and can be disrupted by disease of the host [1]. The microbiome is also well known to be influenced by host sex, age, antibiotics, diet, and other environmental factors [2,3,4]. In fact, the microbiome can become altered in such a way that diversity decreases and pathogenic bacteria dominate over commensal bacteria, also known as the pathobiome state [5]. This dysbiotic state can lead to perpetuation of a disease state and prolonged recovery in addition to downstream consequences of inflammation and immune system dysregulation [5].

### 1.1. Maintenance of the Intestinal Barrier

The intestine consists of multiple layers including the mucosa, submucosa, muscularis propria, and serosa [6]. The mucosa is of particular interest in terms of the microbiome and includes the epithelium, lamina propria, and muscular mucosa [6]. The inner most epithelial layer is held together by complex junctions [7]. From the most superficial, these include tight junctions, adherens junctions, and desmosomes [7]. Tight junctions involve interactions between multiple proteins such as claudins, zonula occludens 1, occludin, and F-actin [7]. Adherens junctions are formed by E-cadherin, alpha-catenin 1, beta-catenin, p120 catenin, and F-actin [7]. Finally, desmoglein, desmocollin, desmoplakin, and keratin filaments make up desmosome junctions [7]. Together, these prevent intestinal contents from translocating outside the gastrointestinal tract but can be affected by various disease states.

### 1.2. Role in the Immune System and Inflammation

The microbiome has a complex relationship with the immune system which is mediated by multiple pathways. The epithelial layer contains enterocytes, goblet cells which secret mucous, and Paneth cells which produce antimicrobial peptides and immunomodulatory proteins [7,8]. While the epithelial layer contains some lymphocytes, other immune cells such as macrophages and dendritic cells exist mainly in the in the lamina propria [9]. Microbiota utilize pathogen-associated molecular patterns (PAMPs) to interact with immune cells which express pattern recognition receptors [10]. These PAMPs affect downstream immune-modulating effects, which are outside the scope of this review, but contribute to either immune regulation during homeostasis or dysregulation in states of disease [10]. Finally, there is an intricate relationship between the gut microbiome and adaptive immune responses [11,12]. Within the intestine, commensal bacteria interact with adaptive immune cells to promote maintenance of the intestinal barrier [11]. These connections are complex and are explored in other works [11,12].

### 1.3. Microbiome Characterization

The microbiome can be described in terms of diversity as well as microbial composition. Alpha-diversity describes community richness and can be represented by a variety of indices. Common measurements of alpha-diversity include the Chao1 index which represents the number of different microbial species present, whereas the Shannon index accounts for both the richness and evenness of species in the microbiome. Operational taxonomic units (OTU) represent a collection of 16S rRNA sequences, whereas the abundance-based coverage estimators (ACE) index is a measure of species richness sensitive to rare OTUs. Beta-diversity is a measurement of community dissimilarities between ecosystems (inter-individual variability) and can be assessed with algorithms such as the Bray–Curtis dissimilarity algorithm and permutation-based multivariate analyses of variance concerning a matrix of pairwise distance to partition the inter-group and intra-group distance. The makeup or taxonomy of the microbiome is categorized in terms of phylum, class, order, family, and genus by relative abundance. This can be further analyzed with algorithms such as linear discriminant analysis effect size (LEfSe), which determines unique bacterial taxa driving specific group-specific differences, or ALDEx2 (ANOVA-like differential expression) and ANCOM (analysis of composition of microbiomes) which utilize different methods to identify unique microbial features in groups [13,14,15]. More recently, metagenomics has emerged as a method to allow for a more in-depth analysis of the microbiome compared to other methods [16,17]. Together, these methods allow for in-depth study and characterization of the intestinal microbiome.

## 2. Trauma 

### 2.1. Hemorrhagic Shock

Initial studies investigating the emergence of intestinal dysbiosis after injury concerned hemorrhagic shock alone (Figure 1). One study in New Zealand rabbits subjected to hemorrhagic shock demonstrated dramatic changes in the intestinal microbiome acutely at 24, 36, 48 h, and one week later as evidenced by acute changes in beta-diversity within 24 h of hemorrhagic shock and increased alpha-diversity which persisted for days afterward [18]. Similarly, a murine model of hemorrhagic shock showed discrete differences in microbial composition after trauma with decreases in commensal bacteria such as *Bacteroidetes* [19]. While it is surprising that these studies had an increase in alpha-diversity after hemorrhagic shock, it is possible that this is related to low injury severity although this remains to be studied. Yang et al. utilized a murine model of hemorrhagic shock and resuscitation to demonstrate increased ileal tissue permeability four hours afterward [20]. Studies of hemorrhagic shock in animals further demonstrated the role of toll-like receptor 5 (TLR-5) as a mediator of the breakdown of the mucosal intestinal barrier in mice [21]. Another group showed that mitochondrial dynamin-related protein is involved in regulation of the intestinal barrier of the colon after hemorrhagic shock in mice [19]. Together, these studies demonstrated the effects of hemorrhagic shock alone on intestinal dysbiosis and gut barrier integrity, thus highlighting the potential for further investigation into microbiome alterations after other injuries.

### 2.2. Traumatic Brain Injury

The effects of traumatic brain injury (TBI) on dysbiosis constitute perhaps the most well studied of all isolated injuries; to understand its effects on the intestinal microbiome, several preclinical and human studies have been conducted (Figure 1). Preclinical studies of the fecal microbiome acutely after traumatic brain injury demonstrate both diversity and microbial composition changes 24 h postinjury compared to controls [22,23,24]. A detailed listing of preclinical studies of microbiota and intestinal barrier changes after TBI can be found in Table 1. 

Multiple preclinical studies have established that traumatic brain injury induces intestinal dysbiosis. Yang et al. evaluated the fecal microbiome over the course of two weeks in male mice after traumatic brain injury and found an increase in alpha-diversity at days three and seven and shifts in beta-diversity over the course of two weeks compared to controls, with distinct microbial composition changes [24]. Hou et al. showed a decline in alpha-diversity at 3 days postinjury in TBI of male rats, beta-diversity differences at days 3 and 7, and microbial composition changes [25]. However, Bao et al., in a model of TBI in male mice, did not identify differences in alpha- or beta- diversity seven days postinjury, although they noted microbial composition changes [26]. Nicholson et al. evaluated changes in the microbiome over time at 2 hours, 24 hours, 3 days and 7 days after TBI and found an acute decrease in alpha-diversity by day 3, beta-diversity shifts at days 1 and 3, and changes in microbial composition [27]. This group concluded that these changes all resolved one week after TBI [27]. However, another group found persistently low alpha-diversity, changes in beta-diversity, and microbiome composition differences seven days after TBI [28]. Similarly, a murine model of TBI in males showed a sharp decline in the Chao1 index one hour after injury which persisted over the course of one week, along with beta-diversity changes and microbial composition alterations [23]. Another study of male rats post-TBI demonstrated microbiome perturbations in terms of alpha-diversity at 3 days which resolved after 1 month, but persistent differences in beta-diversity at 4, 30, and 60 days after injury were found [29]. Even longer-term studies (as long as seven months) demonstrated changes in beta-diversity and microbial composition between one week, one month, and seven months after TBI [30]. Together, this confirms how isolated traumatic brain injury affects microbiome diversity and composition acutely and also long term. 

Few studies have investigated the microbiome at other sites in the intestine after traumatic brain injury. Only 24 h after injury, Wang et al. found that the caecum microbiota in mice had a drastic drop in alpha-diversity, beta-diversity shifts, and also significant microbial composition changes after TBI, with high levels of *Eubacteriumi* and *Roseburia* [31]. Wen et al. also found that the caecum microbiota was altered after traumatic brain injury in mice at multiple time points over the course of one week in terms of diversity and composition [32]. Investigation of the jejunal microbiome in rats subjected to repetitive closed head injuries demonstrated a sharp decline in alpha-diversity at 6 h which persisted for 48 h and as long as 30 days postinjury and, similarly, changes in beta-diversity were found at these time points [33]. In the same study, they demonstrated unique microbiome compositions after TBI, with a decrease in abundance of butyrate-producing bacterial families [33]. An additional study of repetitive TBI in male mice showed a decline in caecum microbiota alpha-diversity at 45 days after injury and a unique makeup of microbial composition at both 45 and 90 days [34]. Male mice subjected to TBI experienced beta-diversity shifts and an increase in *Peptococcaceae* and a decrease in *Prevotellaceae* abundance 72 h postinjury [35]. Thus, more proximal sites in the intestine also demonstrate similar microbiome alterations.

Age, in the setting of TBI, has also been shown to affect dysbiosis as well. One study showed that older mice subjected to TBI had greater changes in disease-associated microbial species compared to younger mice counterparts [36]. Another group performed repetitive TBI on adolescent and adult male and female rats, showing that while there were no differences between age groups in terms of diversity or taxonomy at 17 or 30 days postinjury, pre-injury depletion of the microbiome resulted in a postinjury microbiome with a pathogenic proinflammatory profile, with decreases in commensal bacteria such as *Bacteroides* and *Bifidobacterium*, which was worse in adolescents [37]. This group also found that adolescent females were less resilient and had worse dysbiosis than males subjected to repetitive TBI after thirty days [37]. Thus, additional studies are warranted to expand on the role of age and dysbiosis, along with recovery of the microbiome, after TBI. 

TBI has also been shown to affect the integrity of the intestinal barrier. Ma et al. studied a model of TBI in mice, demonstrating colonic injury along with decreased expression of the tight-junction protein claudin-1 and an increase in paracellular flux 28 days postinjury [38]. Another group evaluated mice at multiple time points over the course of one week and found a decrease in the tight-junction expression of occludin in the colon by day three and decreased expression of claudin-1 and claudin-2 in the colon after one day and the small intestine by day three [23]. Similarly, Sgro et al. subjected rats to TBI and found a decrease in occludin expression in the small intestine at 30 days postinjury [37]. Zhang et al. showed jejunal injury only 2 hours after TBI and hemorrhagic shock in rats, with evidence of capsase-1-positive cells and pyroptotic cells which resolved by day 30 [39]. Evaluation of the small intestine after TBI in male mice demonstrated villous atrophy, decreased lysozyme presence from Paneth cells, and increased caspase-3 levels indicative of apoptosis [24]. Male Sprague-Dawley rats subjected to TBI also showed elevated levels of fluorescein isothiocyanate (FITC)-labeled dextran and lipopolysaccharide (LPS) in plasma at one week and seven months after injury [40]. Another model of TBI in male rats showed decreased intestinal expression of claudin and occludin along with permeability of FITC-dextran to plasma at days 3 and 7 after injury [25]. Some studies have shown that gut-derived metabolites can have neurotoxic effects [31,41]. This evidence shows how traumatic brain injury can affect the intestinal barrier’s integrity.

**Table 1 microorganisms-11-01990-t001:** Major preclinical studies of traumatic brain injury and its effects on the fecal microbiome and/or intestinal barrier. Arrows indicate a significant change (increase or decrease) and ↔ indicates no significant change compared to uninjured controls; we only selected microbial composition changes shown. TBI—traumatic brain injury; HS—hemorrhagic shock. TBI^†^—open; TBI^§^—closed head injury; NS—not studied.

	Author and Year	Species	Trauma Model	Time Point(s) Studied Postinjury	Fecal Microbiome Diversity and Composition Changes	Intestinal Barrier Effects
Age matters: microbiome depletion prior to repeat mild traumatic brain injury differentially alters microbial composition and function in adolescent and adult rats	Sgro et al., 2022 [33]	Male and Female Sprague-Dawley Rats	TBI^§^ (repetitive)	17 and 30 days	↔ Alpha-diversity (30 d) ↔ Beta-diversity (30 d) ↔ Microbial composition (30 d)	↔ Tight-junction protein expression in small intestine (30 d) ↓ Occludin expression in small intestine (30 d)
Traumatic brain injury induces gastrointestinal dysfunction and dysbiosis of gut microbiota accompanied by alterations in bile acid profile	You et al., 2022 [19]	Male C57BL/6J Mice	TBI^†^	1 h, 6 h, 1, 3, and 7 days	↓ Alpha-diversity (all time points) Beta-diversity differences (1 d, 3 d, 7 d) ↓ *Firmicutes* (6 h) ↑ *Bacteroidetes, Proteobacteria* (6 h) Changes in genus composition over time	↔ ZO-1 expression in colon and small intestine (1 d, 3 d) ↓ Occludin expression in colon (3 d) ↓ Claudin-1 expression in small intestine (1 d, 3 d), colon (1 d) ↓ Claudin-2 expression in small intestine (3 d), colon (1 d)
Translocation and dissemination of gut bacteria after severe traumatic brain injury	Yang et al., 2022 [20]	Male C57BL/6 Mice	TBI^†^	3 h, 1, 3, 7, and 14 days	↑ Alpha-diversity (3 d, 7 d) Beta-diversity differences ↓ *Firmicutes, Lactobacillus, Akkermansia, Roseburia* (7 d) ↑ *Bacteroidetes, Streptococcus, Lachnoclostridium* (7 d)	↑ Villous atrophy in small intestine (1 d, 3 d, 7 d, 14 d) ↓ Lysozyme from Paneth cells in small intestine (3 d) ↑ Caspase-3 in small intestine (1 d, 3 d, 7 d, 14 d)
Oral administration of brain protein combined with probiotics induces immune tolerance through the tryptophan pathway	Hou et al., 2021 [21]	Male Sprague-Dawley Rats	TBI^†^	3 and 7 days	↓ Alpha-diversity (3 d) Beta-diversity differences ↑ *Akkermansia* (3 d) ↓ *Lactobacillus* (3 d, 7 d)*, Parabacteroides* (7 d)	↓ Claudin in colon, small intestine (3 d, 7 d) ↓ Occludin in colon, small intestine (3 d, 7 d) ↑ Plasma fluorescein isothiocyanate-dextran (3 d, 7 d)
Sustained dysbiosis and decreased fecal short-chain fatty acids after traumatic brain injury and impact on neurologic outcome	Opevemi et al., 2021 [41]	Male C57BL6/J Mice	TBI^†^	3 h, 1, 3, 7, 14, and 28 days	↓ Alpha-diversity (7 d, 14 d, 28 d) Beta-diversity differences ↑ *Verrucomicrobiaceae, Erysipelotrichaceae* ↓ *Lachnospiraceae, Ruminococcaceae, Bacteroidaceae*	NS
Effects of traumatic brain injury on the gut microbiota composition and serum amino acid profile in rats	Taraskina et al., 2022 [24]	Male Wistar Rats	TBI^†^	7 days	↓ Alpha-diversity Beta-diversity differences ↓ *Bacteroidetes* ↑ *Rikenellaceae*, *Prevotellaceae*, *Lactobacillus*, *Turicibacter*, *Helicobacter*	NS
Traumatic brain injury in mice induces acute bacterial dysbiosis within the fecal microbiome	Treangen et al., 2018 [18]	Male C57BL/6J Mice	TBI^†^	24 h	↓ *Lactobacillus* ↑ *Marvinbryantia*, *Clostridiales*	NS
Moderate traumatic brain injury alters the gastrointestinal microbiome in a time-dependent manner	Nicholson et al., 2019 [23]	Male Sprague-Dawley Rats	TBI^†^	2 h, 1, 3, and 7 days	↓ Alpha-diversity (3 d) Beta-diversity differences (1 d, 3 d) ↓ *Firmicutes* (1 d, 3 d), *Lachnospiraceae* (2 h, 3 d), *Mogibacteriaceaei* (1 d, 3 d) ↑ *Bacteroidaceae* (1 d), *Verrucomicrobia* (1 d), *Enterobacteriaceae* (3 d)	NS
Differential fecal microbiome dysbiosis after equivalent traumatic brain injury in aged versus young adult mice	Davis 4th et al., 2021 [32]	Male C57BL/6 Mice	TBI^†^	1, 7, and 28 days	↔ Beta-diversity over time ↑ *Clostridium clocleatum, Anaerostipes, Lactobacillus, Coprococcus*	NS
Susceptibility to epilepsy after traumatic brain injury is associated with preexistent gut microbiome profile	Medel-Matus et al., 2022 [26]	Male and Female Sprague-Dawley Rats	TBI^†^	7 days, 1 month, and 7 months	↔ Alpha-diversity over time Beta-diversity differences over time ↓ *Lachnospiraceae* (7 d, 1 mo, 7 mo), *Lactobaccillus* (1 mo) ↑ *Ruminiclostridium, Pseudomonas* (7 d), *Bacteroides* (1 mo), *Parabacteroides* (7 mo)	NS
Traumatic brain injury alters the gut-derived serotonergic system and associated peripheral organs	Mercado et al., 2022 [38]	Male C57BL/6J Mice	TBI^†^	1, 3, and 7 days	↓ *Clostridium leptum* (7 d) ↑ *Clostridium scindens* (7 d)	NS
Acute gut microbiome changes after traumatic brain injury are associated with chronic deficits in decision making and impulsivity in male rats	Frankot et al., 2023 [25]	Male Long-Evans Rats	TBI^†^	3, 30, and 60 days	↓ Alpha-diversity (3 d v 30 d) Beta-diversity differences over time ↓ *Firmicutes, Bacteroides*	NS
An integrated analysis of gut microbiota and the brain transcriptome reveals host–gut microbiota interactions following traumatic brain injury	Bao et al., 2023 [22]	Male C57BL/6 Mice	TBI^†^	7 days	↔ Alpha-diversity ↔ Beta-diversity ↑ *Actinobacteria, Bifidobacteriales, Bifidobacteriaceae, Bifidobacterium*	NS
Bidirectional brain–gut interactions and chronic pathological changes after traumatic brain injury in mice	Ma et al., 2017 [34]	Male C57BL/6 Mice	TBI^†^	24 h, 28 days	NS	↑ Colonic injury (28 d) ↔ Jejunal injury (24 h, 28 d) ↓ Transepithelial electrical resistance in jejunum (24 h) ↑ Paracellular flux in colon (28 d) ↔ ZO-1 in colon (28 d) ↔ Occludin expression in colon (28 d) ↓ Claudin-1 in colon (28 d) ↔ Claudin-2 in colon (28 d)
CORM-3 exerts a neuroprotective effect in a rodent model of traumatic brain injury via the bidirectional gut–brain interactions	Zhang et al., 2021 [35]	Male Sprague-Dawley Rats	TBI^†^ & HS	24 h, 30 days	NS	Jejunal histological changes at 24 h postinjury characterized by cleaved caspase-1-positive cells, pyroptotic cells
Disruption of the intestinal barrier and endotoxemia after traumatic brain injury: implications for post-traumatic epilepsy	Mazarati et al., 2021 [36]	Male Sprague-Dawley Rats	TBI^†^	1 week, 7 months	NS	↑ Plasma lipopolysaccharide (1 week, 7 month) ↑ Plasma fluorescein isothiocyanate-labeled dextran (1 week, 7 month)

Given the findings of microbiome changes after TBI, some studies sought to investigate potential connections between dysbiosis and bone formation, post-traumatic epilepsy, neurologic outcomes, and post-traumatic stress disorder. One group investigated dysbiosis after TBI and found that pre-injury microbiome-depleted rats had disruption of femur growth, with different aspects of growth irregularities between males and females [42]. Another study showed that post-traumatic epilepsy can be stratified based on microbial composition [30]. One murine study showed acutely decreased expression of neuronal markers in the duodenum and colonic tissue one day after TBI, which may affect intestinal peristalsis [43]. In addition, another study which subjected normal mice or antibiotic-treated (gut microbiota-depleted) mice to TBI showed that those who were given antibiotics before or after injury had reduced neuronal death and microglial cell count compared to untreated counterparts after three days [44]. On the contrary, Celorrio et al. showed that antibiotic-induced dysbiosis prior to TBI resulted in increased neuronal loss, reduced microglial density, and decreased T-lymphocyte infiltration at the injured site compared to untreated counterparts [45]. Mazarati et al. found correlated disruption of the intestinal barrier after TBI with post-traumatic motor dysfunction [40]. Short-chain fatty acid (SCFA) administration to male mice after TBI resulted in improved spatial learning postinjury [46]. Houlden et al. evaluated male mice after different TBI severities and identified a correlation between cecal dysbiosis and injury severity 72 h after injury [35]. Finally, traumatic brain injury and post-traumatic stress disorder have been shown to have similar neuropsychiatric symptoms and underlying pathophysiology [47]. Recent studies have demonstrated how gut microbiome alterations may facilitate the development of PTSD [48,49,50]. While additional discussions into the underlying mechanisms behind the gut–brain axis are outside the scope of this review, they are further discussed elsewhere [51,52,53,54,55].

Parallel studies of TBI in human subjects have demonstrated intestinal dysbiosis after injury. Mahajan et al. evaluated the fecal microbiome composition of 101 patients with moderate or severe TBI admitted to an intensive care unit and showed a decrease in *Enterobacteraceae* at days three and seven postinjury compared to controls [56]. Hou et al. collected stool samples from 24 patients with moderate or severe TBI and identified increased alpha-diversity along with shifts in beta-diversity and depletion of *Bifidobacterium* and *Faecalibacterium* compared to healthy controls [25]. Urban et al. evaluated the fecal microbiome of 22 patients at an average of 20 years after moderate or severe traumatic brain injury residing in permanent care facilities and identified an increased Shannon index, differences in beta-diversity, and an abundance of *Firmicutes*, *Actinobacteria*, and *Verrucomicrobia* compared to healthy controls [57]. One of the largest human studies of the fecal microbiome after TBI included 34 patients with moderate or severe TBI and compared them to 79 patients without TBI and to 297 patients with no TBI or mild TBI [58]. Brenner et al. collected samples from patients with moderate/severe TBI an average of 28 years after injury and discovered no differences in alpha-diversity, beta-diversity, or microbial composition between these patients and those with no TBI or mild TBI, suggesting that dysbiosis after TBI resolves at this later time point [58]. Together, this demonstrates acute changes in the fecal microbiome after head injuries, with differences that may persist 20 years postinjury but may resolve at later time points. Additional longer-term studies to identify the temporal changes in the microbiome after isolated TBI are warranted to better understand this recovery of dysbiosis.

### 2.3. Spinal Cord Injury

Isolated spinal cord injury (SCI) has also been studied to evaluate gut dysbiosis postinjury, although mainly in female animal subjects (Figure 1). Du et al. evaluated female mice subjected to thoracic spinal cord crush injury at varying levels and showed that a proximal spinal cord injury (T4) resulted in a shift in beta-diversity and different microbial composition with more pathogenic organs compared to a more distal spinal cord crush injury (T10) 21 days postinjury, suggesting a potential involvement of sympathetic preganglionic neurons in the development of dysbiosis after SCI [59]. Kigerl et al. performed T9 spinal cord contusion in female mice and analyzed the fecal microbiome at five different time points over the course of 28 days after injury and observed a decrease in *Bacteroidales* and an increase in *Clostridiales* [60]. Female mice subjected to spinal cord injury via contusion at the T8-10 level resulted in decreased alpha-diversity and differences in beta-diversity and microbial composition with increased *Firmicutes* and *Proteobacteria* and decreased abundance of *Bacteroidetes* two weeks after injury compared to sham groups which only underwent laminectomy [61]. Female rats subjected to T9 spinal cord contusion demonstrated beta-diversity shifts along with increased *Bifidobacteriaceae* and *Clostridiaceae* eight weeks after injury, although alpha-diversity metrics were similar between groups at this time point [62]. This same group demonstrated increased expression of proinflammatory factors interleukin-1 beta and interleukin-12 in the small intestines of injured animals after four weeks [62]. A model of contusive spinal cord injury in female pigs at the level of T2 or T10 demonstrated differences in beta-diversity and microbial composition compared to uninjured counterparts; this was most pronounced within two weeks of injury over the eight-week duration of study [63]. A study of T10 spinal cord contusion in mice showed beta-diversity differences between injured mice and controls along with different makeup of microbes at seven days postinjury [64]. This highlights the influence of spinal cord injury alone on both diversity and microbial composition of the intestinal microbiome both acutely and weeks after injury. 

Similar to TBI, SCI recovery is influenced by intestinal dysbiosis. One study of antibiotic-induced intestinal dysbiosis prior to SCI resulted in an exacerbation of neurologic impairment after injury [60]. A study of male and female mice who underwent L5 spinal cord transection and intestinal microbiome depletion with oral antibiotics demonstrated inhibited neuropathic pain [65]. Myers et al. investigated the role of *Pde4b*, which plays a role in neuroinflammation and axonal degeneration, in female mice after SCI and found that mice lacking this gene avoided postinjury dysbiosis and also had improved hindlimb locomotion recovery [66]. These studies suggest a complex relationship between the nervous system and the microbiome which warrants study [67]. 

Human studies have also been performed to evaluate the effects of spinal cord injury on the microbiome, showing alterations in diversity and composition from one month to thirteen years afterward. One study of 23 patients with spinal cord injury showed that the fecal microbiome as early as 1 month (but up to 25 years) postinjury demonstrated a surprising increase in alpha-diversity (OTU, Chao1 index, PD index, Simpson and Shannon indices) but also shifts in beta-diversity and an abundance of the phylum *Actinobacteria* and genera *Lachnospiraceae UCG-008*, *Ruminococcus*, and *Lactobacillus*, among others [68]. Bazzocchi et al. studied 100 patients and evaluated their intestinal microbiome within 60 days of injury and found no differences in alpha-diversity but found shifts in beta-diversity [69]. They also characterized the microbial composition of these patients and observed an abundance of *Methanobrevibacter, Streptococcus, Enterococus, Klebsiella*, and *Akkermansia* in patients with spinal cord injury and that the severity of dysbiosis varied by lesion level and completeness of spinal cord injury [69]. Zhang et al. evaluated the fecal microbiome of 43 male patients with complete spinal cord injury at least six months postinjury and found a decline in alpha-diversity (Simpson), beta-diversity changes, and also an abundance of genera such as *Bacteroides*, *Blautia*, *Lachnoclostridium*, and *Escherichia-Shigella* after injury compared to healthy controls [70]. Another study of 20 patients with cervical SCI demonstrated a decline in alpha-diversity (Simpson), beta-diversity differences, and an abundance of *Proteobacteria*, *Verrucomicrobia*, *Bacteroides*, *Blautia*, *Esherichia-Shigella*, *Lactobacillus*, and *Akkermansia* at least six months postinjury [71]. Additionally, 11 patients with cervical or thoracic SCI had a fecal microbiome with fewer species, beta-diversity shifts, and persistent microbial composition differences at an average of 1.9 years postinjury [72]. Gungor et al. evaluated the fecal microbiome of 30 patients with SCI at various levels; stool samples were collected from patients, on average, 19.5 months after injury and no differences were found in total DNA bacterial groups between groups [73]. However, this study did find differences in microbial composition of the intestinal microbiome when injured patients were grouped by lower motor neuron (T12 or distal) or upper motor neuron (proximal) injuries [73]. Lin et al. evaluated the microbiome of 23 patients with spinal cord injuries; the majority of patients were male and samples were collected, on average, 11 months postinjury [74]. This study found that injured patients had decreased Chao1 index, OTUs, and ACE; although not statistically significant compared to healthy controls, differences in beta-diversity and microbial composition between groups were found [74]. Li et al. compared differences in the gut microbiome acutely after SCI in seven patients and long term after SCI at a mean of 18 years postinjury in 25 patients and found both an acute increase in alpha- diversity (Chao1 index, observed species) and long-term increase compared to healthy controls along with shifts in beta-diversity [75]. Patients who suffered acute SCI had an abundance of *Sutterella* and *Odoribacter* whereas long-term SCI patients had more *Clostridiales* [75]. Yu et al. studied the intestinal microbiome in 21 patients with complete SCI and 24 patients with incomplete SCI an average of 5.6 months postinjury and found decreased alpha-diversity in both groups compared to healthy controls along with similar shifts in beta-diversity, with complete SCI having a more distinct shift from controls than incomplete SCI [76]. Incomplete SCI patients had abundances of *Lactobacillaceae*, *Lachnospiraceae*, *Eubacterium*, *Clostridium*, and *Sutterella* whereas complete SCI patients’ intestinal microbiomes were dominated by *Coriobacteriaceae*, *Syngergistetes*, *Eubacterium*, and *Cloacibacillus* [76]. Pattanakuhar et al. studied long-term changes in the fecal microbiome of patients with SCI an average of 13 years postinjury and found prominence of *Bacteroides* and *Firmicutes* [77]. In fact, this group demonstrated that the level of SCI was correlated with gut microbiota profiles [77]. These data highlight the importance of level of SCI and completeness of SCI on intestinal dysbiosis. Additional studies are warranted to further investigate changes in the microbiome more acutely postinjury and also over time. 

### 2.4. Multicompartmental Injury

Several animal studies have been conducted to understand how multiple injuries can impact the intestinal microbiome and the intestinal barrier (Figure 1). Table 2 shows detailed summaries of published animal studies which utilized models of multiple injuries and subsequent effects on the fecal microbiome and also the intestinal barrier. Regarding the fecal microbiome, Kelly et al. performed lung contusion and hemorrhagic shock (LCHS) with varying durations of postinjury stress in male rats to replicate intensive care units for either 7 days or 14 days postinjury [78]. This group highlighted the impact of postinjury stress on the intestinal microbiome; they found that chronic daily stress after LCHS resulted in increased alpha-diversity at different time points dependent on the duration of stress [78]. They also revealed how *Bacteroides* levels were affected with the addition or removal of chronic stress [78]. 

More severe models of multiple injuries characterized as polytrauma have demonstrated both acute and long-term dysbiosis in animal studies. Nicholson et al. studied male rats subjected to a polytrauma model of femur fracture, hemorrhagic shock, and crush injuries to the small intestine, liver, and skeletal muscle of an extremity and evaluated the fecal microbiome two hours postinjury [79]. This group found no changes in alpha-diversity but found drastic shifts in beta-diversity with associated changes in microbial composition, with injured rats having abundances of *Lachnospiraceae* [79]. Appiah et al. subjected male mice to a polytrauma model consisting of bilateral chest trauma, closed head injury, femur fracture, and soft tissue injury with or without hemorrhagic shock and found no differences in alpha- or beta-diversity at 280 min postinjury, although this study concerned the microbiome of the cecum [80]. Furthermore, this group correlated increasing alpha-diversity metrics of the cecal microbiome with elevated plasma cytokines such as interleukin-6; they also showed distinct composition of the cecal microbiome after polytrauma with or without hemorrhagic shock [80]. Another study of multiple injuries in male rats consisting of laparotomy, liver and skeletal muscle crush injuries, and femur fracture with hemorrhagic shock illustrated beta-diversity differences and unique microbiome composition changes with increased levels of *Roeburia* and *Enterobactericeae* along with decreased levels of *Rothia* and *Streptococcus*, despite no significant differences in alpha-diversity being found at this early time point [81]. Furthermore, this group correlated changes in alpha-diversity postinjury with mesenteric oxygen concentration [81].

Longer-term animal studies have evaluated intestinal microbiome changes over the course of one week. Our group previously evaluated the microbiome at days three and seven of male rats in a model of polytrauma (unilateral lung contusion, cecectomy, bifemoral pseudofracture to simulate femur fracture, and hemorrhagic shock) with or without chronic stress (PT, PT/CS) and demonstrated a drastic decrease in alpha-diversity three days postinjury, with persistently low alpha-diversity only in rats subjected to multiple injuries with chronic stress [82]. This study also demonstrated shifts in beta-diversity in both PT and PT/CS at days three and seven, with unique microbial composition differences between these groups at each time point [82]. A shorter-term study of the same polytrauma model in both males and females demonstrated that at baseline pre-injury, females had a higher Shannon index (alpha-diversity) than males and after injury, females and males had similar alpha-diversity [83]. This study also showed that males and females subjected to polytrauma demonstrated differences in beta-diversity 2 days postinjury, in addition to unique microbiome compositions [83]. The evidence in these studies shows the impact of polytrauma on the intestinal microbiome and also underscores the need for more studies investigating the role of sex in the development of the pathobiome and the need to uncover the underlying mechanisms behind postinjury dysbiosis.

Other studies have also sought to understand how multiple injuries impact the intestinal barrier postinjury. Rupani et al. studied a model of laparotomy with sham shock or laparotomy with hemorrhagic shock on male rats and evaluated ileal histology along with translocation of orally administered FITC-dextran to serum at multiple time points within 3 h of injury [84]. This group found changes in ileal histology postinjury consistent with villous injury and loss of villous height, decreased mucus layer, and increased enterocyte apoptosis [84]. Wbra et al. studied a model of polytrauma (blunt chest trauma, closed head injury, femur fracture with soft tissue injury) with or without hemorrhagic shock in male mice and showed that two hours after injury, mice subjected to polytrauma with hemorrhagic shock demonstrated decreased expression of the tight-junction protein zonula occludens protein 1 in the ileum and colon [85]. In addition, all experimental groups in this study (hemorrhagic shock, polytrauma, polytrauma with hemorrhagic shock) demonstrated elevated plasma intestinal fatty-acid-binding protein levels, suggestive of increased intestinal permeability [85]. A previously mentioned study of male rats subjected to unilateral lung contusion, cecectomy, hemorrhagic shock, and bifemoral pseudofracture with or without chronic stress resulted in increased plasma occludin levels seven days postinjury after multiple injuries with stress, highlighting the potential influence of chronic stress on increased intestinal permeability [82]. This study also highlighted an increased descending colonic injury in both groups of multiple injuries with or without chronic stress characterized by lamina propria edema and the infiltration of inflammatory cells seven days postinjury [82]. Finally, a study of the same model of polytrauma in both male and female rats showed increased plasma occludin and lipopolysaccharide-binding proteins in both PT and PT/CS groups of males and females, suggestive of increased intestinal permeability, only two days after injury [83]. This same study also showed that males subjected to PT had higher levels of plasma occludin and males subjected to PT/CS had higher plasma LBP than their female counterparts 48 h postinjury [83]. In addition, males showed worse ileum injury after PT/CS than female counterparts at two days postinjury [83]. These data show the influence of multiple injuries on the intestinal barrier and the role that sex may play in this response to injury.

Few studies have been conducted in humans to understand the effects of multiple and severe injuries on the intestinal microbiome. Burmeister et al. studied 67 trauma patients with an average injury severity score (ISS) of 21 and performed a rectal swab on admission and characterized the microbiome acutely postinjury [86]. This group identified that higher alpha-diversity was correlated with increased survival; in addition, shifts in beta-diversity postinjury were correlated with body mass index, sex, length of hospital stay, length of intensive care unit stay, and mortality [86]. This study also correlated decreased *Firmicutes* and blooms of *Prevotella* and *Corynebacterium* in the intestinal microbiome on admission with survival [86]. Howard et al. investigated changes to the microbiome via serial rectal swabs in 12 trauma patients with a median ISS of 27 and found that upon admission and 24 h postinjury, there were no differences in terms of alpha- or beta-diversity between injured patients and healthy controls [87]. Additional samples taken at 72 h postinjury demonstrated shifts in the beta-diversity of trauma patients with associated changes in microbial composition consisting of an abundance of *Clostridiales* and *Enterococcus* and depletion of *Bacteroidales, Fusobacteriales,* and *Verrucomicrobiales* compared to healthy controls [87]. Finally, Nicholson et al. studied 72 patients admitted after severe injury with an average ISS of 21 and evaluated changes in the microbiome with serial samples taken over the course of almost two weeks [88]. These data showed an initial increase in alpha-diversity but then a significant decrease by day five of hospital stay which persisted for almost two weeks and did not recover [88]. Similarly, these subjects had beta-diversity shifts upon admission which persisted and were even found to be correlated with injury severity score [88]. Trauma patients from this study also had a unique microbiome profile characterized by depletion of *Firmicutes* and an abundance of *Proteobacteria* compared to healthy controls at all time points [88]. Importantly, this study showed that patients who received more blood products tended to have a higher alpha-diversity and beta-diversity more similar to healthy controls [88]. These findings illustrate not only the acute changes in the microbiome postinjury but also the persistence of an altered intestinal microbiome postinjury up to almost two weeks after trauma, along with the influence of blood transfusion on the gut microbiome. Additional human studies with subgroup analysis to understand the effects of sex, age, and injury severity on intestinal dysbiosis are warranted.

## 3. Therapeutics

### 3.1. Probiotics, Prebiotics, and other Medications

Regarding the strong evidence demonstrating the emergence of a pathobiome after isolated or multiple injuries, studies have sought to investigate different therapeutics to mitigate this dysbiosis. One study administered quercetin, a polyphenol flavonoid which has been implicated as a prebiotic with neuroprotective effects, after repeated traumatic brain injury in mice and found that it not only improved bacterial abundance in the intestinal microbiome, but it also restored fecal short-chain fatty-acid levels such as acetate, butyrate, and propionate and attenuated histologically evident colonic injury one week postinjury [89]. Yanckello et al. administered inulin, a prebiotic containing carbohydrates that contribute to the growth of bacteria which produce short-chain fatty acids, postinjury in a murine traumatic brain injury model which resulted in an abundance of commensal bacteria, reduced pathogenic bacteria, increased fecal short-chain fatty acids in the cecal microbiome, and improved cerebral blood flow compared to untreated counterparts at five months postinjury [90]. This same group also investigated pre-injury administration of inulin for 2 months in the same murine model and demonstrated a decrease in pathogenic bacteria, an increase in short-chain fatty-acid-producing bacteria, and a different beta diversity than untreated counterparts in cecal contents at 24 h, 1.5 months, and 3 months postinjury [91]. Li et al. administered *Clostridium butyricum*, a butyrate-producing probiotic, for two weeks before and two weeks after traumatic brain injury in mice and showed a reduction in the expression of inflammatory markers in the colon and preservation of colonic occludin expression in treated mice compared to untreated mice and also demonstrated improved neurologic deficits and degeneration, less brain edema, and amelioration of neuronal apoptosis [92]. Ma et al. administered *Lactobacillus acidophilus* after traumatic brain injury in mice for either one, three, or seven days and showed increased alpha-diversity and restoration of microbial composition along with neuroprotective effects [93]. Jing et al. studied a murine model of spinal cord contusion with postinjury administration of melatonin and found decreased intestinal permeability with increased expression of zonulin and occludin in the colon, restored commensal bacteria in the intestinal microbiome, and improved postinjury locomotor testing in treated mice after four weeks [94]. Rong et al. investigated SCI in mice and found that subjects experienced improved synaptic regeneration in addition to preservation of intestinal microbiota diversity 21 days postinjury when administered ursolic acid, a pentacyclic triterpenoid [95]. He et al. administered resveratrol, a phenol, to mice after SCI which restored the intestinal microbiota from postinjury dysbiosis and increased fecal butyrate at seven days [96]. Tian et al. showed in a murine model of hemorrhagic shock and resuscitation that administration of *n*-3 or *n*-6 polyunsaturated fatty acids afterward resulted in improved mucin production and increased goblet cells in the intestines and an abundance of commensal bacteria compared to untreated counterparts only 12 h after injury [97]. Hou et al. studied a model of brain injury in male rats and showed that administration of brain proteins and *Lactobacillus* and *Bifidobacterium* probiotics resulted in elevated alpha-diversity and decreased intestinal permeability compared to untreated counterparts in addition to decreased circulating inflammatory cytokines within two weeks of injury [25]. Thus, these potential therapeutics could be investigated in human studies for efficacy.

Few studies in human subjects have been performed to attempt to meliorate postinjury gut dysbiosis. However, Brenner et al. performed a study of the use of *Limosilactobacillus reuteri* daily for 8 weeks in 16 TBI patients, on average, 10 years postinjury and showed that this probiotic decreased plasma C-reactive protein levels, but that there were no differences in alpha- or beta-diversity or microbial composition compared to those who received a placebo [98]. Further investigation into the ability of probiotics to improve postinjury dysbiosis after isolated injuries or polytrauma are warranted to validate the strong evidence provided in animal studies.

### 3.2. Fecal Microbiota Transplantation

Fecal transplantation from a healthy subject may be another potential option to resolve postinjury dysbiosis and also influence injury recovery or outcomes. One study performed following TBI in mice with subsequent weekly fecal transplantation noted significantly higher alpha-diversity, different beta-diversity, and microbial composition alterations (90). In addition, they found improved neurologic functional testing compared to untreated mice [99]. This same group studied the effects of daily fecal transplantation for four weeks in mice subjected to TBI and found that this resulted in reduced microglial inflammatory gene expression and even mitigated T-cell response postinjury compared to mice who were subject to TBI alone [100]. Du et al. studied TBI in male rats and performed daily fecal transplantation for seven days, showing increased alpha-diversity, shifts in beta-diversity, and microbial composition differences along with improved neurological testing results and serum metabolite differences compared to rats who were subject to TBI alone after eight days [101]. Jing et al. studied SCI in mice and then performed fecal transplantation for four weeks and showed improved alpha-diversity, different microbial composition, and increased colonic expression of zonulin and tight-junction protein occludin and fecal short-chain fatty acids. This group also demonstrated that fecal transplantation after SCI resulted in improved injury recovery and promotion of neuronal axonal regeneration [102]. The same group then showed that fecal transplantation in a murine SCI model also improved neuronal survival and white matter sparing of the spinal cord along with spinal cord ischemia reduction, improved vascular repair, protection against blood–spinal cord barrier disruption, and alleviation of neuroinflammation [103]. Rong et al. utilized a murine model of SCI with subsequent fecal transplantation and also showed increased survival of neurons, microglia, and astrocyte cells after one week [64]. Thus, improvements in dysbiosis postinjury may not only improve the microbiome but may also affect injury recovery.

## 4. Discussion

The intestinal microbiome and intestinal barrier together form a complex system which maintains homeostasis but can be disrupted after trauma, not only in isolated injuries but also after multicompartmental injuries, in terms of diversity and microbial composition. Studies on different therapeutics including probiotics, prebiotics, medications, and even fecal transplantation demonstrate benefits regarding not only restoration of the microbiome but also in injury recovery. Further studies into the role of sex and age in postinjury dysbiosis, fecal metabolomics, underlying mechanisms between injury and the gut microbiome, and the potential role of fecal microbiota transplantation in severe injury should be pursued to improve outcomes after trauma.

## Figures and Tables

**Figure 1 microorganisms-11-01990-f001:**
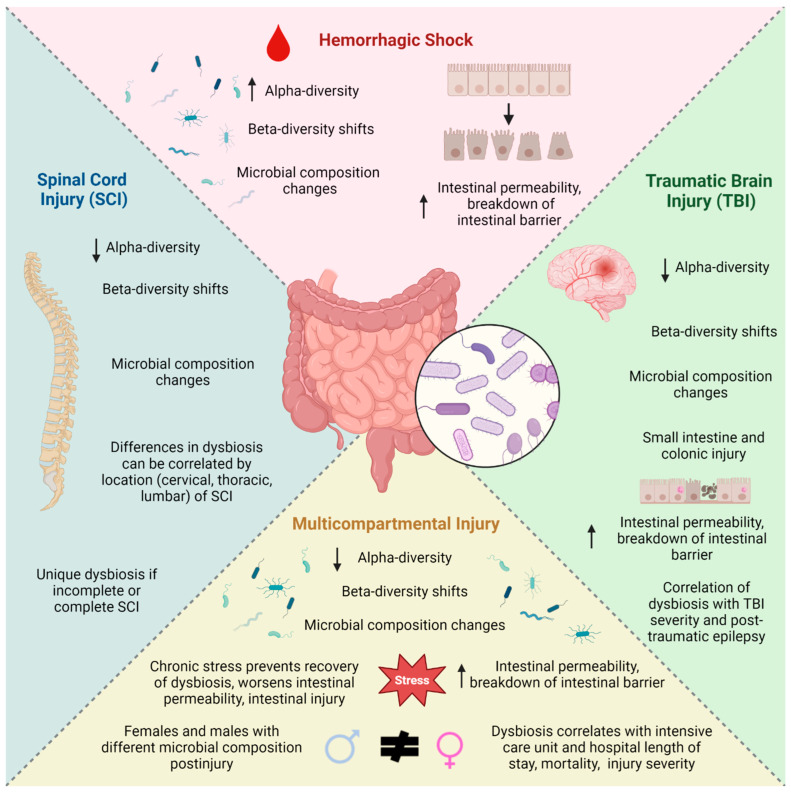
Changes in the microbiome after various injuries. Created with BioRender.com.

**Table 2 microorganisms-11-01990-t002:** Major preclinical studies of multiple injuries and effects on the fecal microbiome and/or intestinal barrier. Arrows indicate a significant change (increase or decrease) and ↔ indicates no significant change compared to uninjured controls; we only selected microbial composition changes shown. T—trauma, HS—hemorrhagic shock, LC—lung contusion, LCHS—lung contusion and hemorrhagic shock, CS—chronic stress, polytrauma (PT), NS—not studied.

Title	Author and Year	Species	Trauma Model	Time Point(s) Studied Postinjury	Fecal Microbiome Diversity and Composition Changes	Intestinal Barrier Effects
Multicompartmental traumatic injury and the microbiome: shift to a pathobiome	Munley et al., 2023 [73]	Male Sprague-Dawley Rats	Lung contusion, hemorrhagic shock, laparotomy with cecectomy, and bifemoral pseudofracture (PT), or PT with CS (PT/CS)	3, 7 days	↓ Alpha-diversity (PT, 3 d) ↓ Alpha-diversity (PT/CS, 3 d, 7 d) Beta-diversity differences (3 d, 7 d) ↑ *Enterococcus, Bacteroides, Parabacteroides* (PT, 3 d) ↑ *Bacteroides* (PT/CS, 7 d)	↑ Plasma occludin (PT/CS, 7 d) ↑ Colonic injury (inflammatory cell infiltrates, lamina propria edema) (PT and PT/CS, 7 d)
Multicompartmental Traumatic Injury Induces Sex-Specific Alterations in the Gut Microbiome	Munley et al., 2023 [74]	Male and Female Sprague-Dawley Rats	Lung contusion, hemorrhagic shock, laparotomy with cecectomy, and bifemoral pseudofracture (PT), or PT with CS	2 days	↓ Alpha-diversity (PT, PT/CS both sexes) Beta-diversity differences (PT, PT/CS both sexes) ↑ *Blautia, Bacteroides* (PT, both sexes) ↑ *Parasutterella, Frisingicoccus* (PT/CS, both sexes)	↑ Plasma occludin (PT, PT/CS both sexes) ↑ Plasma lipopolysaccharide-binding protein (PT, PT/CS both sexes) ↑ Ileum injury (PT/CS, males) ↑ Colon injury (PT/CS, females)
Polytrauma independent of therapeutic intervention alters the gut microbiome	Nicholson et al., 2018 [70]	Male Sprague-Dawley Rats	Laparotomy with small intestine and liver crush injuries, femur fracture, skeletal muscle crush, HS (PT)	2 h	↔ Alpha-diversity Beta-diversity differences ↑ *Deferribacteres*, *Lachnospiraceae*, *Mogibacteriaceae*	NS
Whole blood resuscitation restores intestinal perfusion and influences gut microbiome diversity	Yracheta et al., 2021 [72]	Male Sprague-Dawley Rats	Laparotomy with liver crush injury, skeletal muscle crush injury, femur fracture (PT), HS	2 h	↔ Alpha-diversity Beta-diversity differences ↑ *Roseburia*, *Enterobacteriaceae* ↓ *Rothia*, *Streptococcus*	NS
Stress-related changes in the gut microbiome after trauma	Kelly et al., 2021 [69]	Male Sprague-Dawley Rats	LCHS, LCHS and 7 d CS (LCHS/CS 7/7), or LCHS and 14 d CS (LCHS/CS 14)	3, 7, 14 days	↔ Alpha-diversity (LCHS, all time points) ↑ Alpha-diversity (LCHS/CS 7/7, 3 d, 7 d) ↑ Alpha-diversity (LCHS/CS 14, 7 d, 14 d) ↔ Beta-diversity (7 d, 14 d) ↑ *Ruminococcaceae* (3 d), *Bacteroides* (7 d), *Lactobacillus* (14 d) (LCHS/CS 7/7) ↑ *Clostridium* (3 d), *Bacteroides* (7 d), *Lachnospiraceae* (14 d) (LCHS/CS 14)	NS
Relationship between disruption of the unstirred mucus layer and intestinal restitution in loss of gut barrier function after traumatic hemorrhagic shock	Rupani et al., 2007 [75]	Male Sprague-Dawley Rats	Laparotomy and sham shock (T/SS), or laparotomy and HS (T/HS)	0, 30, 60, 180 min	NS	↑ Villous injury in ileum (T/HS, 60 min) ↓ Villous height in ileum (T/HS, all time points) ↑ Enterocyte apoptosis in ileum (T/HS, 30, 60 min) ↓ Mucus layer in ileum (T/HS) ↑ Serum fluorescein isothiocyanate-dextran 4 (T/HS, 0, 60 and 180 min)
Remote intestinal injury early after experimental polytrauma and hemorrhagic shock	Wrba et al., 2019 [76]	Male C57BL/6 Mice	Hemorrhagic shock (HS), or bilateral chest trauma, closed head injury, femur fracture with soft tissue injury (PT), or PT and HS (PT/HS)	2 h	NS	↑ Plasma intestinal fatty-acid-binding protein (HS, PT, PT/HS) ↓ Zonula occludens protein 1 ileum, colon (PT/HS)

## Data Availability

No new data were created or analyzed in this study. Data sharing is not applicable to this article.

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
