# Peer review of "The Intestinal Microbiome after Traumatic Injury"

_microorganisms, 2023, doi:10.3390/microorganisms11081990_

Round 1

Reviewer 1 Report

In this review, Munley and colleagues discuss the link between alterations of the intestinal microbiome and traumatic injury.  The manuscript is well-written and well-organized. However, I have only one minor point that could improve this manuscript.

-       There is a high comorbidity between TBI and PTSD. TBI and PTSD are characterized by common neuropsychiatric symptoms including anxiety, irritability, insomnia, and cognitive deficits. Moreover, overlapping pathophysiological changes have been observed in preclinical models of TBI and PTSD (PMID: 29017388). In this context, the Authors might want to add the most recent translational studies (PMID: 36400332; PMID: 35789769; PMID: 36970450). showing gut microbiota alterations after traumatic stress. The possible overlapping between these alterations and those observed after traumatic injuries should be discussed.

-       The Authors should discuss the neurotoxic effects of gut-derived metabolites produced after trauma.

Author Response

In this review, Munley and colleagues discuss the link between alterations of the intestinal microbiome and traumatic injury. The manuscript is well-written and well-organized. However, I have only one minor point that could improve this manuscript.

We thank the reviewer for their kind comments regarding our manuscript.

  1. There is a high comorbidity between TBI and PTSD. TBI and PTSD are characterized by common neuropsychiatric symptoms including anxiety, irritability, insomnia, and cognitive deficits. Moreover, overlapping pathophysiological changes have been observed in preclinical models of TBI and PTSD (PMID: 29017388). In this context, the Authors might want to add the most recent translational studies (PMID: 36400332; PMID 35789769; PMID: 36970450) showing gut microbiota alterations after traumatic stress. The possible overlapping between these alterations and those observed after traumatic injuries should be discussed.

We have added discussion of the effects of traumatic stress on the gut microbiome to the traumatic brain injury section including these references.

  1. The authors should discuss the neurotoxic effects of gut-derived metabolites produced after trauma.

We have added a statement regarding the neurotoxic effects of gut-derived metabolites produced after trauma to the traumatic brain injury section.

Reviewer 2 Report

The intestinal microbiome's critical role in host immune function and homeostasis has led to studies examining its dysregulation in patients and models of traumatic injuries. Trauma results in changes in gut microbiome diversity and composition, transitioning to a pathobiome state. Additionally, sex has been identified as a biological variable influencing these alterations. Therapeutics like fecal transplantation are used to address microbiome changes and support recovery post-injury. This review summarizes post-injury gut microbiome alterations, proposed mechanisms, and future research directions, which is interesting and timely for the readers. This review is well written and structured. I only have some minor comments.

Line 61. Role in the Immune System and Inflammation. The adaptive immune responses were closely related to gut microbiota as well. It would be good to see more statements about these relationships.

Line 73. Microbiome Characterization. In addition to 16S rRNA sequencing, metagenomics is also an important tool to investigate the microbial community, and it could provide more details and depth sequences.

Line 102. Higher alpha-diversity was observed in Hemorrhagic Shock, while lower diversity was observed in TBI, SCI, and Multicompartment injury, the possible explanations?

Discussion section. In addition to focus on the role of sex and age in postinjury dysbiosis, etc. Another direction is may be to uncover the mechanism of those microbiome regulating the pathogenesis or development of traumatic injuries.

Author Response

The intestinal microbiome’s critical role in host immune function and homeostasis has led to studies examining its dysregulation in patients and models of traumatic injuries. Trauma results in changes in gut microbiome diversity and composition, transitioning to a pathobiome state. Additionally, sex has been identified as a biological variable influencing these alterations. Therapeutics like fecal transplantation are used to address microbiome changes and support recovery post-injury. This review summarizes post-injury gut microbiome alterations, proposed mechanisms, and future research directions, which is interesting and timely for the readers. This review is well written and structured. I only have some minor comments.

We thank the reviewer for their kind comments regarding our manuscript.

  1. Line 61. Role in the Immune System and Inflammation. The adaptive immune responses were closely related to gut microbiota as well. It would be good to see more statements about these relationships.

A statement about the relationship between the gut microbiome and adaptive immune responses has been added to this section.

  1. Line 71. Microbiome Characterization. In addition to 16S rRNA sequencing, metagenomics is also an important tool to investigate the microbial community, and it could provide more details and depth sequences.

A statement about metagenomics has been added to this section.

  1. Line 102. Higher alpha-diversity was observed in Hemorrhagic Shock, while lower diversity was observed in TBI, SCI and Multicompartmental Injury, the possible explanations?

Discussion with potential explanations of higher alpha-diversity after hemorrhagic shock alone has been added to the Hemorrhagic Shock section.

  1. Discussion section. In addition to focus on the role of sex and age in postinjury dysbiosis, etc. Another direction is may be to uncover the mechanism of those microbiome regulating the pathogenesis or development of traumatic injuries.

A statement about the role of the microbiome in the development of traumatic injuries has been added to the discussion.